# Significant changes of the choroid in patients with ocular ischemic syndrome and symptomatic carotid artery stenosis

Hae Min Kang[1]*, Jeong Hoon Choi[2], Hyoung Jun Koh[3], Sung Chul Lee[3]

1 Department of Ophthalmology, Catholic Kwandong University College of Medicine, International St. Mary's Hospital, Incheon, Republic of Korea, 2 Choikang Eye Clinic, Seoul, Republic of Korea, 3 Department of Ophthalmology, Yonsei University College of Medicine, Seoul, Republic of Korea

* liebe05@naver.com

## Abstract

### Purpose

To evaluated the changes in choroidal vasculature in patients with ocular ischemic syndrome (OIS) and in the ipsilateral eyes of patients with symptomatic carotid artery stenosis (CAS)

### Method

A total of 50 patients (15 patients with OIS, 10 patients with symptomatic CAS, 25 patients of age-and sex-matched control group) were included, and the medical records were retrospectively reviewed. The mean subfoveal choroidal thickness (SFCT) of each eye was measured, and binary images of the choroid were evaluated to compare the mean choroidal area and the luminal area.

### Results

The mean SFCT was 170.5±75.3 μm in the eyes with OIS, 154.8±62.9 μm in the ipsilateral eyes with symptomatic CAS, and 277.5±73.2 μm in the right eyes of the control group patients ($P<0.001$). The mean choroidal area was 494,478.6±181,846.2 μm2 in the eyes with OIS, 453,750.0±196,725.8 μm2 in the ipsilateral eyes with symptomatic CAS, and 720,520±281,319.5 μm2 in the control group eyes ($P = 0.036$). The mean luminal area was 333,185.7±112,665.9 μm2 in the eyes with OIS, 313,983.3±132,032.1 μm2 in the ipsilateral eyes with symptomatic CAS, and 480,325.0±185,112.6 μm2 in the control group eyes ($P = 0.046$). The mean SFCT, mean choroidal area, and mean luminal area were significantly smaller in the eyes with OIS ($P = 0.017$, $P = 0.005$, and $P = 0.004$, respectively), and those with symptomatic CAS ($P = 0.020$, $P = 0.016$, and $P = 0.021$, respectively) than in the unaffected contralateral eyes. There were no significant differences between the eyes in the control group ($P = 0.984$, $P = 284$, and $P = 0.413$, respectively).

### Conclusion

The mean SFCT, mean choroidal area, and mean luminal area were significantly thinner in the eyes with OIS and the ipsilateral eyes with symptomatic CAS, compared with the control

**Data Availability Statement:** TData are available upon request due to restrictions imposed by the Institutional Review Board of International St. Mary's Hospital involving confidential data. Data

are available from the approval Institutional Data Access / Ethics Committee (contact via Je Hoon Park, ceccil@ish.ac.kr) for researchers who meet the criteria for access to confidential data.

**Funding:** This work was supported by a National Research Foundation of Korea (NRF) grant funded by the Korea government (MSIT) (No. 2018R1C1B5085620). The funder had no role in study design, data collection and analysis, decision to publish, or preparation of the manuscript.

**Competing interests:** The authors have declared that no competing interests exist.

group eyes. The eyes with OIS and those with symptomatic CAS had significantly thinner SFCT, and smaller choroidal area and luminal area than the unaffected contralateral eyes. Choroid may reflect the vascular status of the carotid artery, indicated by choroidal thinning and decreasing choroidal area, especially luminal area.

## Introduction

The retina has a dual blood supply; the blood supply for the inner retina is derived from retinal vasculature and the outer retina is supplied solely by the choroidal vasculature [1]. The retina, especially the photoreceptors, is extremely metabolically active; >90% of the oxygen supplied to the retina is consumed by the photoreceptors [2]. To maintain the oxygen supply to the outer retina, especially to the photoreceptors, the choroid maintains a high blood flow [2]. This blood flow is the highest of any tissue in the body per unit tissue weight; it is nearly ten-fold higher than the blood flow to the brain [2]. Thus, stenosis or occlusion of the internal carotid artery (ICA) or the common carotid artery (CCA) may lead to impaired ocular blood circulation because the ophthalmic artery is the first intradural branch of the ICA. Ocular manifestations such transient monocular blindness or ocular arterial occlusive disease can sometimes precede the signs of cerebrovascular disease (e.g., ischemic stroke) [3–8].

Ocular ischemic syndrome (OIS) is caused by chronic ocular hypoperfusion due to a >90% stenosis or complete occlusion of the CCA or ICA [9,10]. OIS usually occurs in the mid-sixties age group, and men are affected at a greater rate than women due to higher incidence of cardiovascular and atherosclerotic diseases [11–14]. The prevalence of OIS is not precisely known, but one study estimated that OIS occurs at a rate of 7.5 cases per million persons every year [15].

OIS is associated with poor vision at the time of diagnosis, leading to permanent visual loss [11,13,14,16]. In addition to visual deterioration, OIS may be preceding signs for predict devastating cerebrovascular or cardiovascular disease, which leads to increased morbidity and mortality [7,8,12,17]. Thus, patients with OIS should be recommended a thorough systemic evaluation to minimize the risk of morbidity and mortality.

With the advancements in various multi-modal imaging analyses, we previously compared mean subfoveal choroidal thickness (SFCT) values between OIS-affected eyes and the non-affected contralateral eyes in three patients with ipsilateral OIS [18]. In our case study, the mean SFCT was thinner in the OIS-affected eyes compared with the non-affected contralateral eyes. Our case study results suggested that impaired choroidal circulation secondary to carotid artery occlusive disease may lead to choroidal thinning in patients with OIS. Following our case study, subsequent studies found significant choroidal thinning in eyes with OIS, which supports our findings [19,20]. A recent study investigated central choroidal thickness (CCT) in patients with ipsilateral ICA stenosis >65% and amaurosis fugax (AF), but no other ocular manifestations [21]. The study found that the mean CCT value of the ipsilateral eyes with severe CAS was significantly lower than that of the control group. This result suggested that the choroidal thinning may occur before the retinal manifestations in patients with OIS.

In this study, we investigated whether there were any changes in the choroid among patients with OIS, patients with ipsilateral symptomatic carotid artery stenosis (CAS) and no obvious ocular manifestations, and the normal control group. We also investigated whether there were any changes in choroidal structure in patients with OIS and in those with ipsilateral symptomatic CAS.

## Methods

### Study population

This retrospective study was performed at the Catholic Kwandong University College of Medicine, International St. Mary's Hospital. The study protocol was approved by the Institutional Review Board of International St. Mary's Hospital, Catholic Kwandong University and adhered to all tenets of the Declaration of Helsinki. The requirement for informed consent from each patient was waived due to the retrospective nature of the investigation. This waiver was also approved by the Institutional Review Board.

We retrospectively reviewed medical records for the September 2014 to October 2018 period and selected patients for inclusion in the study groups. Group 1 was defined as the patients who underwent ophthalmologic evaluation and had a diagnosis of treatment-naïve unilateral OIS. Group 2 was defined as the patients with ipsilateral symptomatic CAS with an opposite ICA stenosis <40% and no other concomitant ocular disease. Symptomatic CAS was defined as intracranial or extracranial stenosis of the ICA leading to symptoms of transient monocular blindness or AF, transient ischemic attack (TIA), or ischemic stroke ipsilateral to the lesion [22]. Group 3 was the age- and sex-matched normal control group; these patients had no concomitant ocular disease. The exclusion criteria were: 1) eyes with any sign of pathologic myopia such as fundus changes indicative of pathologic myopia, including lacquer cracks, atrophic patches, or chorioretinal atrophy; 2) previous panretinal photocoagulation for any concomitant chorioretinal disease; 3) concomitant other ocular disease such as glaucoma and any sign of retinal vascular diseases such as diabetic retinopathy; and 4) any intraocular surgery such as pars plana vitrectomy, scleral buckling, or intravitreal injections 3 months before the diagnosis of OIS.

All patients with OIS and with symptomatic CAS underwent head and neck computed tomography examinations. The examinations were performed SIEMENS Somatom Definition flash (128 x 2 channels) (SIEMENS medical, Germany). The stenosis ratio was calculated using the North American Symptomatic Carotid Endarterectomy Trial stenosis grading method [23,24].

### Ophthalmologic examinations of the study population

Ophthalmologic examinations (e.g., a slit lamp examination, an intraocular pressure measurement using a non-contact tonometer, and a fundus examination) were performed for each patient. The refractive error value was measured for each eye using an autorefractor, and the result was converted to spherical equivalents [diopters (D)]. Multi-modal imaging studies included fluorescein angiography (FA), fundus auto fluorescence, and spectral domain optical coherence tomography (SD OCT) (Spectralis; Heidelberg Engineering) with an enhanced depth imaging (EDI) modality. FA was performed using the Heidelberg Retina Angiograph system (HRA-2; Heidelberg Engineering, Heidelberg, Germany), with a confocal scanning laser ophthalmoscope.

### Choroidal evaluation by spectral domain optical coherence tomography

The choroid of each patient was evaluated using SD OCT with the EDI modality. EDI OCT imaging was performed by positioning the objective lens of the Spectralis OCT scanner close enough to invert the image. At least two good-quality horizontal and vertical scans across the fovea were obtained for each eye. We also obtained 37 horizontal macular scans of 30 x 15 degrees through the center of the fovea. Choroidal thickness was defined as the distance from the outer border of the hyperreflective line, corresponding to the retinal pigment epithelium

(RPE) perpendicular to the chorio-scleral interface. Using the digital calipers provided by the Heidelberg Spectralis OCT software, the SFCT was measured at the subfoveal region in each trans-sectional image of the horizontal and verticalmacular scan and then averaged. The SFCT was measured by two independent observers (HMK and HJC) who were blinded to the clinical data of each patient.

After measurement of the SFCT, the subfoveal choroidal images were used for further binarization. Binarization of the subfoveal choroidal area in the SD OCT image was performed using a modified Niblack method as previously described [25,26]. The OCT image was analyzed using ImageJ software (version 1.51j8; provided in the public domain by the National Institutes of Health, Bethesda, MD, USA; http://imagej.nih.gov/ij/). The examined area was selected to be 1500 μm wide, with margins 750 μm nasal and 750 μm temporal to the fovea. It extended vertically from the RPE and to the chorioscleral border, and the choroidal area was set with the Image J ROI manager. Then, three choroidal vessels with lumens larger than 100 μm were randomly selected by the Oval selection Tool on the ImageJ tool bar, and the average reflectivity of these areas was evaluated. The average brightness was set as the minimum value to minimize the noise in the OCT image. Then, the image was converted to 8 bits and adjusted by the auto local threshold of Niblack. The binarized image was reconverted to an RGB image, and the luminal area was determined using the threshold tool. After the data for the distance of each pixel were added, the total choroidal area, luminal area, and stromal area were automatically calculated. The light pixels were defined as the stromal choroid and the dark pixels were defined as the luminal area.

## Statistical analysis

IBM SPSS Statistics Version 22.0 software for Windows (IBM Corporation, Somers, NY, USA) was used for statistical analyses. Mauchly's test of sphericity and Kolmogorov-Smirnov analyses were used to confirm statistical validity. Baseline characteristics (e.g., age at the time of diagnosis and sex) were evaluated. The presence of hypertension (HTN), diabetes mellitus (DM), cardiovascular disease such as myocardial infarct (MI), and cerebrovascular disease such as TIA or ischemic stroke was investigated. The mean SFCT, mean choroidal area, mean luminal area, and mean stromal area that were obtained from the SD OCT images were also evaluated. Non-parametric analyses were used due to the relatively small study population number. Between-group comparisons, those with OIS and symptomatic CAS were grouped as Group 1, and those in the normal control group were grouped as Group 2. Then, statistical analyses between two groups were performed using the Mann-Whitney U test for continuous variables and the chi-square test for categorical variables. The Wilcoxon sign rank test was used for the intra-person comparisons of the values for mean SFCT, mean choroidal area, mean luminal area, and mean interstitial area. The Kruskal-Wallis test was used for comparisons of mean SFCT, mean choroidal area, mean luminal area, and mean interstitial area among the three groups. Results with a $P < 0.05$ were considered statistically significant.

## Results

### Baseline characteristics

A total of 50 patients were included in this study. The mean age at the time of ophthalmologic evaluation was 68.7±9.7 (range, 47–86) years. Among the 25 patients with OIS and symptomatic CAS, 17 (68.0%) patients were male. Fifteen (30%) patients had OIS; 10 (20.0%) had symptomatic CAS, but not OIS; and 25 (50.0%) patients were in the age-and sex-matched control group.

## Comparison of baseline characteristics between the patients with carotid artery occlusive diseases and those of control group

We subdivided the study population into two groups: Group 1 as the patients with carotid artery occlusive diseases, in other words, those with OIS and symptomatic CAS, and Group 2 as the age-and sex-matched normal population. There were no statistically significant differences in mean age at the time of ophthalmologic evaluation ($P = 0.975$) or sex ($P = 0.990$) among the two groups (groups 1 and 2). There was no significant difference in the prevalence of HTN ($P = 0.167$) and DM ($P = 0.109$), however, the prevalence of cardiovascular diseases such as MI and cerebrovascular diseases such as TIA were significantly higher in group 1 than group 2 ($P = 0.019$ and $P = 0.015$, respectively). The results of the comparisons are presented in more detail in Table 1.

## Comparison of choroidal characteristics among the study population: the patients with ocular ischemic syndrome, those with symptomatic carotid artery stenosis, and the control group

We compared the mean SFCT, the mean choroidal area, the mean luminal area, and the stromal area among three groups: the patients with OIS, those with symptomatic CAS, and age- and sex-matched control group. The mean SFCT was 170.5±75.3 μm (68.0 μm– 289.5 μm; median, 163.5 μm) in the eyes with OIS, 154.8±62.9 μm (76.5 μm– 280.0 μm; median, 156.3 μm) in the ipsilateral eyes with symptomatic CAS, and 277.5±73.2 μm (122.0 μm– 285.0 μm; median, 278.0 μm) in the right eye of the control group. The detailed values are shown in Table 2. The mean SFCT was significantly thinner in the eyes with OIS and those with ipsilateral symptomatic CAS than those of control group ($P<0.001$), whereas there was no statistical difference among the contralateral eyes among three groups ($P = 0.871$).

**Table 1. Baseline characteristics of the study population.**

| | Group 1 | | Group 2 | P value |
|---|---|---|---|---|
| | Patients with ocular ischemic syndrome | Patients with symptomatic carotid artery stenosis | Control group (N = 25) | |
| | (N = 15) | (N = 10) | | |
| Age (years) | 69.3±10.0 | 67.6±9.6 | 67.6±10.5 | 0.975* |
| | | 68.7±9.7 | | |
| Sex (M, %) | 10 (66.7%) | 7 (70.0%) | 17 (68.0%) | 0.999† |
| | | 17 (68.0%) | | |
| Axial lengths (mm) | 22.1±0.6 | 21.9±0.9 | 22.5±0.8 | 0.630* |
| | | 22.0±0.8 | | |
| Hypertension | 12 (80.0%) | 7 (70.0%) | 14 (56.0%) | 0.167† |
| | | 19 (76.0%) | | |
| Diabetes mellitus | 8 (53.3%) | 4 (40.0%) | 5 (20.0%) | 0.109† |
| | | 12 (48.0%) | | |
| Cardiovascular disease within 3 months | 5 (33.3%) | 1 (10.0%) | 0 | 0.019† |
| | | 6 (24.0%) | | |
| Cerebrovascular disease within 3 months | 3 (20.0%) | 10 (100.0%) | 0 | 0.015† |
| | | 13 (52.0%) | | |

Statistical analysis was performed by *Mann-Whitney U test for the continuous variables, and †Chi-square test for the categorical variables. Results with $P < 0.05$ were considered statistically significant.

**Table 2. The mean subfoveal choroidal thickness (SFCT), the mean choroidal area, the mean luminal area, and the mean interstitial among the study population: the patients with ocular ischemic syndrome (OIS), the patents with symptomatic carotid artery stenosis (CAS), but no OIS, and the control group.**

|  | Patients with OIS (Group 1, N = 15) | | Patients with symptomatic CAS (Group 2, N = 10) | | Control group (N = 25) | |
|---|---|---|---|---|---|---|
|  | OIS-affected eyes | Contralateral eyes | Ipsilateral eyes with symptomatic CAS | Contralateral eyes | Right | Left |
| The mean SFCT (µm) | 170.5±75.3 | 254.7±88.6 | 154.8±62.9 | 246.1±86.5 | 277.5±73.2 | 279.3±72.1 |
| The mean choroidal area (µm2) | 494,478.6±181,846.2 | 689,850.0±263,525.7 | 453,750.0±196,725.8 | 654,012.5±281,774.7 | 720,520±281,319.5 | 672,035.0±224,916.9 |
| The mean luminal area (µm2) | 333,185.7±112,665.9 | 467,278.6±175,940.7 | 313,983.3±132,032.1 | 437,912.5±179,557.9 | 480,325.0±185,112.6 | 458,075.0±164,950.0 |
| The mean stromalarea (µm2) | 162,685.7±72,695.1 | 222,571.4±95,811.0 | 139,800.0±72,695.1 | 249,212.5±165,286.7 | 236,160.0±97,169.3 | 223,210.0±73,931.7 |

In the eyes with OIS, the mean choroidal area was 494,478.6±181,846.2µm2 (219,403.8 µm2–577,401.3 µm2) and the mean luminal area was 333,185.7±112,665.9µm2 (135,503.8 µm2–404,891.3 µm2). In the eyes with symptomatic CAS, the mean choroidal area was 453,750.0±196,725.8 µm2 (193,508.1 µm2–523,910.3 µm2) and the mean luminal area was 431,437.5±231,259.1 µm2 (141,900.5 µm2–400,618.3 µm2). In the control group, the mean choroidal area of the right eye was 720,520±281,319.5 µm2 (245,209.1 µm2–1,225,601.8 µm2; median, 751,600.8 µm2) and the mean luminal area was 480,325.0±185,112.6 µm2 (154,803.5 µm2–748,409.3 µm2; median, 500,000.8 µm2). The mean choroidal area, the mean luminal area, and the mean stromal area were significantly smaller in the eyes with OIS and those with symptomatic CAS than those of the control group ($P = 0.036$, $P = 0.046$, and $P = 0.037$, respectively). However, there were no significant differences among the unaffected contralatearal eyes of three groups in the mean choroidal area, the mean luminal area, and the mean stromal area ($P = 0.540$, $P = 0.780$, and $P = 0.817$, respectively).

## Intra-personal comparison of choroidal characteristics in the study population

We also investigated intra-personal differences of choroidal characteristics among three groups: the patients with OIS, those with symptomatic CAS, and age-and sex-matched control group. The mean SFCT was significantly thinner in the eyes with OIS than unaffected contra-lateral eyes ($P = 0.017$). The mean SFCT was also significantly smaller in the ipsilateral eyes with symptomatic CAS than unaffected contralateral eyes ($P = 0.020$). However, therwe was no significant difference between two eyes in the control group ($P = 0.994$).

The mean choroidal area, the mean luminal area, and the mean stromal area were significantly smaller in the eyes with OIS ($P = 0.005$, $P = 0.004$, and $P = 0.038$, respectively) and those with ipsilateral symptomatic CAS ($P = 0.016$, $P = 0.021$, and $P = 0.014$, respectively) than the contralateral eyes. There were no statistically significant differences in the mean choroidal area ($P = 0.282$), mean luminal area ($P = 0.413$), or mean stromal area ($P = 0.504$) between the right and the left eyes in the control group.

The representative figures are shown in Figs 1–3, respectively.

## Discussion

Recently, several studies have been performed to investigate choroidal changes in patients with OIS using multi-modal imaging analysis. One study evaluated the diagnostic value of laser speckle flowgraphy for the diagnosis of OIS and its efficacy for the detection of disparities in fundus blood flow [27]. Some study findings suggested there is significant choroidal thinning and decreased choroidal volume in eyes with OIS compared with unaffected contralateral eyes

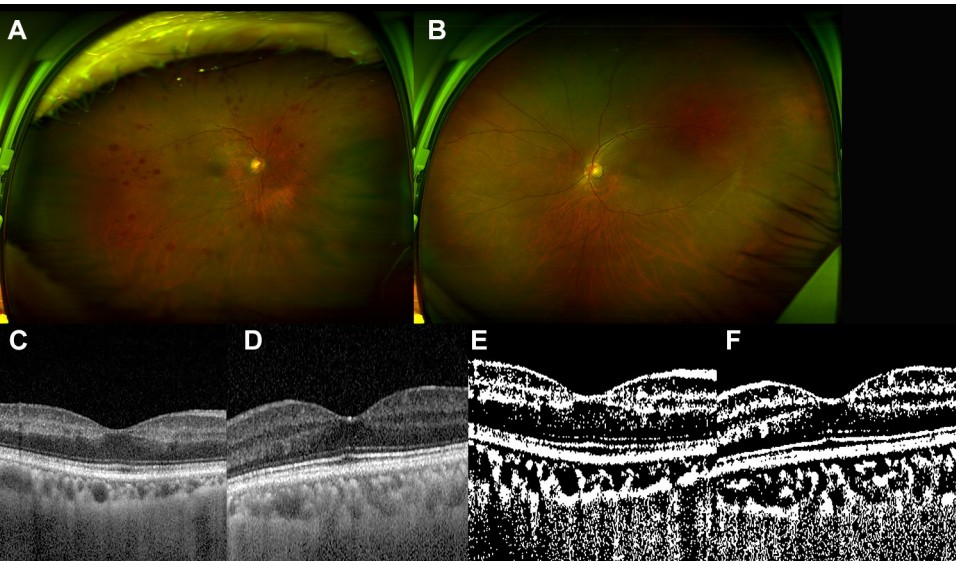

**Fig 1. The 63-year-old male patient with ocular ischemic syndrome in the right eye.** He had past history of hypertension and myocardial infaction. (A and B) (A) Multiple blot retinal hemorrhages were noted in the right eye, whereas (B) there was no significant finding in the left eye. (C and D) Spectral domain optical coherence tomography showed inner retinal thinning in the right eye (C), whereas normal foveal contour in the left eye (D). The mean subfoveal choroidal thickness by spectral optical coherence tomography was 99.0 $\mu$m in the right eye and 201.5 $\mu$m in the left eye. (E and F) After binarization, (E) the choroidal area was 316,100.2 $\mu$m2 and the luminal area was 212,300.2 $\mu$m2 in the right eye. (F) The choroidal area was 516,100.3 $\mu$m2 in the left eye and the luminal area was 348,400.5 in the left eye.

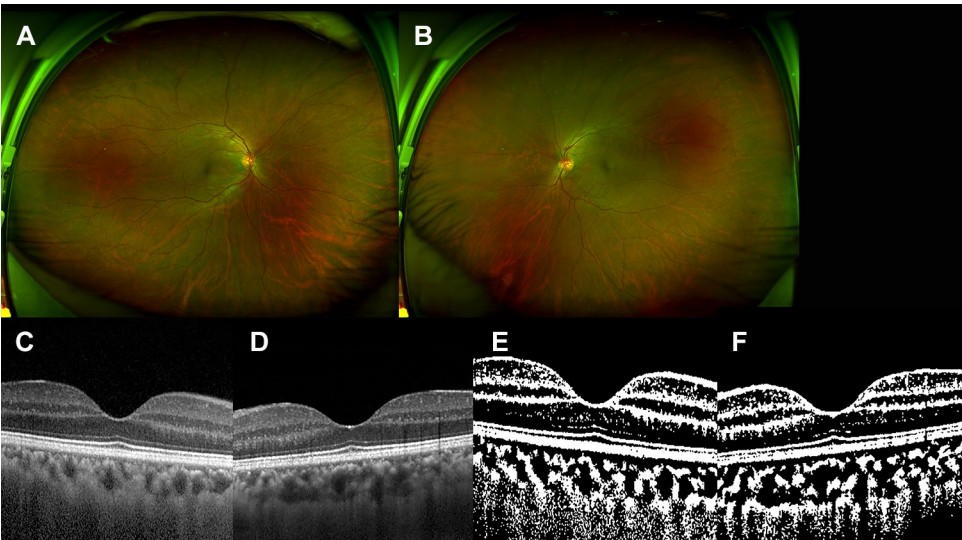

**Fig 2. The 53-year-old male patient with symptomatic carotid artery stenosis.** He had past history of hypertension and cerebrovascular attack, and severe stenosis of right carotid artery at the time of presentation. (A and B) There was no significant finding in the right eye (A) and the left eye (B). (C and D) Spectral domain optical coherence tomography showed no significant sign in both eyes (C, right eye and D, left eye). The mean subfoveal choroidal thickness by spectral domain optical coherence tomography was 187.0 $\mu$m in the right eye and 244.0 $\mu$m in the left eye. (E and F) After binarization, (E) the choroidal area was 503,200.0 $\mu$m2 and the luminal area was 316,100.5 $\mu$m2 in the right eye. (F) The choroidal area was 690,300.5 $\mu$m2 and the luminal area was 471,000.0 in the left eye.

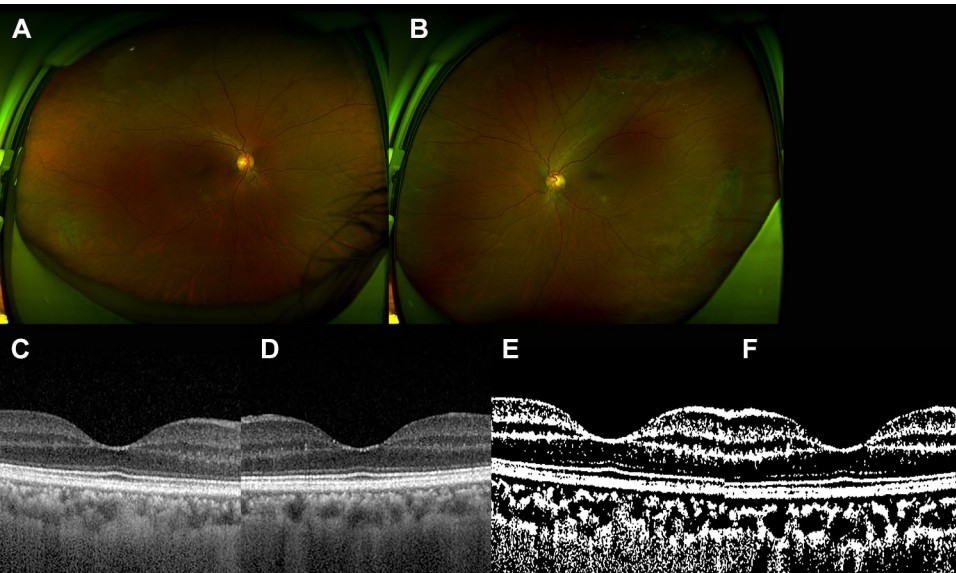

**Fig 3. The 61-year-old female patient who underwent uncomplicated cataract surgery in both eyes.** She had past history of hypertension. (A and B) There was no significant sign in the right eye (A) and lattice degeneration in the left eye (B). (C and D) There was no significant finding in each eye by spectral domain optical coherence tomography (C, right eye, and D, left eye). The mean subfoveal choroidal thickness was 207.5 $\mu$m in the right eye and 195.5 $\mu$m in the left eye. (E and F) After binarization, (E) the choroidal area was 516,100.0 $\mu$m2 and the luminal area was 322,600.0 $\mu$m2 in the right. (F) The choroidal area was 496,800.0 $\mu$m2 and the luminal area was 354,800.0 $\mu$m2 in the left eye.

[18–20]. One study found statistically significant choroidal thinning in patients with severe CAS [21]. Taken together, these study results suggest that choroidal changes may precede the development of OIS.

Here, we compared the choroidal changes in the patients with OIS, those with symptomatic CAS but no definite OIS, and the control group. The mean SFCT value was significantly thinner in the eyes with OIS and those with ipsilateral CAS, compared with the control group. In addition, the mean choroidal area, the mean luminal area, and the mean stromal area were significantly smaller in the eyes with OIS and those with symptomatic CAS than those in the control group. Intra-personal comparison also showed significantly reduced choroidal parameters in the eyes with OIS and those with symptomatic CAS than unaffected contralateral eyes.

The results of this study support previous findings of significant choroidal thinning in eyes with OIS [18–20] and in those with ipsilateral severe CAS [21]. Because symptomatic CAS is itself associated with chronic carotid vascular insufficiency, leading to ischemic cerebrovascular disease, we could assume that significant choroidal thinning can occur in these patients, along with choroidal vascular insufficiency. Our findings suggest that if carotid artery vascular insufficiency progresses to a certain extent, choroidal vascular disturbance and subsequent choroidal thinning may develop before complete obstruction of the carotid artery or >90% of CAS.

We further investigated the changes in choroidal structures using binarization of subfoveal SD OCT images. The results indicated that the mean choroidal area, the mean luminal area, and the mean stromal area were significantly smaller in the patients with OIS and those with ipsilateral symptomatic CAS, compared with the control group. These results indicated that the choroidal vascular insufficiency seems to affect the whole choroidal structures, both choroidal vessels and stroma. In general, 90% CAS reduces ipsilateral central retinal artery perfusion pressure by approximately 50% [9]. In patients with OIS, patchy or delayed choroidal

filling is the most specific angiographic finding, and prolonged retinal arteriovenous time is also common [9,11,20,28,29]. We could speculate that choroidal vascular insufficiency associated with CAS leads to a decrease in, or atrophy of, choroidal vasculature, and a resulting smaller luminal area. Along with choroidal vascular changes, the stroma also can be affected by choroidal vascular insufficiency, resulting in ischemic atrophy and subsequent shrinkage.

Subsequent intra-person comparisons revealed that there were statistically significant differences in the values for mean SFCT, mean choroidal area, and mean luminal area between the eyes with OIS and the unaffected contralateral eyes. The mean SFCT, the mean choroidal area, the mean luminal area, and the mean stromal area were significantly decreased in the eyes with OIS than unaffected contralateral eyes. The eyes with ipsilateral symptomatic CAS also showed similar tendency, whereas the eyes in the control group did not. Based on our findings, we could speculate that choroidal vascular structures are sensitively affected by the vascular status of the carotid arteries, even before ocular manifestations. We also could speculate that symptomatic CAS is associated with ischemic cerebrovascular disease and that there can be a vascular insufficiency in ocular blood flow that has not yet reached the threshold for clinically apparent OIS. This sub-threshold ocular vascular insufficiency associated with symptomatic CAS may lead to choroidal thinning and decreasing choroidal area, affecting both luminal area and stromal area. When the ocular vascular insufficiency reaches a 'thresh-hold', OIS develops and the choroidal changes such as choroidal thinning and a smaller choroidal area (especially in the luminal area) are exacerbated.

We assume that the choroidal changes that occur along with carotid artery obstructive disease progress chronically and gradually. During this chronic and gradual course, significant choroidal change may become prominent at some stages (e.g., development of ischemic cerebrovascular disease), and end-stage disease seems to be the development of OIS. Our results also suggested that significant changes in the choroidal structures and significant choroidal thinning may develop before the apparent ophthalmologic findings of OIS. Thus, ophthalmologic screening of patients with carotid artery occlusive disease, and especially of those with symptomatic CAS, may be effective for detecting and preventing OIS and related causes of visual loss in these patients.

The use of interventions for carotid artery occlusive diseases may improve choroidal blood flow in patients with OIS [30–33]. This improvement in carotid artery blood flow using an intervention may not completely reverse the OIS, but it may prevent further visual loss associated with OIS and improve vision in some patients without neovascular glaucoma. Because the patients with symptomatic CAS usually admit to the hospitals with neurologic symptoms, they undergo systemic evaluation and management when they are diagnosed. Thus, the patients with symptomatic CAS, choroidal blood flow can be improved by systemic management including carotid artery interventions, and their eyes may not progress to OIS. Timely intervention for carotid artery occlusive disease may prevent further vascular insufficiency to the choroid and lead to the prevention of OIS. The results of further studies on choroidal changes after carotid artery intervention in patients with symptomatic CAS may support this hypothesis. If so, the visual loss associated with carotid artery occlusive disease could be prevented.

This study had some limitations, which were associated with the use of a retrospective design. Because ophthalmologic screening is not a routinely-performed examination for patients with symptomatic CAS, the number of patients available for the study was relatively small. Angiographic data was not acquired for every patient, especially for the patients with symptomatic CAS. We also lacked data on the collateral circulation between ECA and ICA. There may be less vascular insufficiency in the eye if there is collateral circulation between the ECA and ICA. Further prospective study is warranted to overcome the effects of these limitations and support our findings.

Although there were some limitations to the study, this is the first comprehensive analysis investigating choroidal changes in patients with OIS and in the ipsilateral eyes with symptomatic CAS. The choroid was significantly thinner in the eyes with OIS and in the ipsilateral eyes with symptomatic CAS, compared with those of the control group. Binary SD OCT images revealed significantly smaller choroidal area, luminal area, and stromal area in the eyes with OIS and in the ipsilateral eyes with symptomatic CAS, compared with those of the control group. The results of the intra-person comparisons indicated that the choroidal changes were significantly changes in the eyes with OIS and those ipsilateral eyes with symptomatic CAS than unaffected contralateral eyes.

In conclusion, carotid artery occlusive disease may be associated with choroidal changes, especially choroidal thinning and smaller choroidal area, both luminal area and stromal area. Ophthalmologic screening and appropriate carotid artery intervention may be helpful in patients with carotid artery occlusive disease and prevent further progression to OIS and associated visual deterioration.

## Author Contributions

**Conceptualization:** Hae Min Kang, Jeong Hoon Choi, Hyoung Jun Koh, Sung Chul Lee.

**Data curation:** Hae Min Kang.

**Formal analysis:** Hae Min Kang, Jeong Hoon Choi, Sung Chul Lee.

**Funding acquisition:** Hae Min Kang.

**Investigation:** Hae Min Kang, Jeong Hoon Choi, Hyoung Jun Koh.

**Methodology:** Hae Min Kang, Jeong Hoon Choi, Hyoung Jun Koh, Sung Chul Lee.

**Project administration:** Hae Min Kang, Hyoung Jun Koh, Sung Chul Lee.

**Resources:** Hae Min Kang.

**Software:** Hae Min Kang.

**Supervision:** Hae Min Kang, Hyoung Jun Koh, Sung Chul Lee.

**Validation:** Hae Min Kang, Jeong Hoon Choi, Hyoung Jun Koh, Sung Chul Lee.

**Visualization:** Hae Min Kang.

**Writing – original draft:** Hae Min Kang, Jeong Hoon Choi.

**Writing – review & editing:** Hae Min Kang, Hyoung Jun Koh, Sung Chul Lee.

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
