## [Decision Letter · Decision Letter 0]

30 Sep 2019

PONE-D-19-22020

Significant changes of the choroid in patients with ocular ischemic syndrome and symptomatic carotid artery stenosis

PLOS ONE

Dear Prof Kang,

Thank you for submitting your manuscript to PLOS ONE. After careful consideration, we feel that it has merit but does not fully meet PLOS ONE’s publication criteria as it currently stands. Therefore, we invite you to submit a revised version of the manuscript that addresses the points raised during the review process.

The manuscript is potentially interesting for the journal and it may be reconsidered provided the authors are willing to address reviewers' concerns.

We would appreciate receiving your revised manuscript by Nov 14 2019 11:59PM. To enhance the reproducibility of your results, we recommend that if applicable you deposit your laboratory protocols in protocols.io, where a protocol can be assigned its own identifier (DOI) such that it can be cited independently in the future. For instructions see: http://journals.plos.org/plosone/s/submission-guidelines#loc-laboratory-protocols

We look forward to receiving your revised manuscript.

Kind regards,

Prof. Raffaele Serra, M.D., Ph.D

Academic Editor

PLOS ONE

**Journal Requirements:**

**Additional Editor Comments (if provided):**

The manuscript is potentially interesting and may be reconsidered provided the authors are willing to address reviewers' concerns.

**Comments to the Author**

1. Is the manuscript technically sound, and do the data support the conclusions?

Reviewer #1: Yes

2. Has the statistical analysis been performed appropriately and rigorously? 

Reviewer #1: Yes

3. Have the authors made all data underlying the findings in their manuscript fully available?

Reviewer #1: No

4. Is the manuscript presented in an intelligible fashion and written in standard English?

Reviewer #1: Yes

5. Review Comments to the Author

Reviewer #1: The authors present a retrospective review evaluating choroidal characteristics among three groups of patients -- those with ocular ischemic syndrome, those with severe carotid artery stenosis, and a control group. They show that multiple choroidal measures, including subfoveal choroidal thickness, choroidal area, luminal area, and stromal area, are decreased with ocular ischemic disease and carotid artery stenosis. This is interesting data and worthy of publication however I suggest that you revise your manuscript to simplify your presentation of the data. As is, you essentially present your data twice -- once with OIS and CAS combined and then again with them separated. I think it would be simpler to just choose one analysis. Both analyses fit your conclusion that ischemic disease, either OIS or carotid artery stenosis, lead to changes in the choroid.

6. PLOS authors have the option to publish the peer review history of their article (what does this mean?). If published, this will include your full peer review and any attached files.

Reviewer #1: No

---

## [Author Response · Author response to Decision Letter 0]

3 Oct 2019

Reviewer #1: The authors present a retrospective review evaluating choroidal characteristics among three groups of patients -- those with ocular ischemic syndrome, those with severe carotid artery stenosis, and a control group. They show that multiple choroidal measures, including subfoveal choroidal thickness, choroidal area, luminal area, and stromal area, are decreased with ocular ischemic disease and carotid artery stenosis. This is interesting data and worthy of publication however I suggest that you revise your manuscript to simplify your presentation of the data. As is, you essentially present your data twice -- once with OIS and CAS combined and then again with them separated. I think it would be simpler to just choose one analysis. Both analyses fit your conclusion that ischemic disease, either OIS or carotid artery stenosis, lead to changes in the choroid.

Answer: We authors appreciate the comment, and also agree with the reviewer’s comment, and made simpler as recommendation. We chose to present the characteristics of choroidal structures among three groups (OIS, symptomatic CAS, and the control group), and removed Table 2. In addition, we authors decided to simplify the Table 2 (previously Table 3), and moved the Table 2 to the paragraph of ‘Comparison of choroidal characteristics among the study population: the patients with ocular ischemic syndrome, those with symptomatic carotid artery stenosis, and the control group’. We authors also modified the next paragraph ‘Intra-personal comparison of choroidal characteristics in the study population’. Thank you again for your valuable comment to the manuscript.

---

## [Editor Report · Decision Letter 1]

9 Oct 2019

Significant changes of the choroid in patients with ocular ischemic syndrome and symptomatic carotid artery stenosis

PONE-D-19-22020R1

Dear Dr. Kang,

We are pleased to inform you that your manuscript has been judged scientifically suitable for publication and will be formally accepted for publication once it complies with all outstanding technical requirements.

With kind regards,

Prof. Raffaele Serra, M.D., Ph.D

Academic Editor

PLOS ONE

Additional Editor Comments (optional):

amended manuscript is acceptable
---

## [Editor Report · Acceptance letter]

14 Oct 2019

PONE-D-19-22020R1 

Significant changes of the choroid in patients with ocular ischemic syndrome and symptomatic carotid artery stenosis 

Dear Dr. Kang:

I am pleased to inform you that your manuscript has been deemed suitable for publication in PLOS ONE. Congratulations! Your manuscript is now with our production department. 

With kind regards,

on behalf of

Prof. Raffaele Serra 

Academic Editor

PLOS ONE